# Transcatheter Mitral Valve Repair or Replacement: Competitive or Complementary?

**DOI:** 10.3390/jcm11123377

**Published:** 2022-06-13

**Authors:** Zhang Xiling, Thomas Puehler, Lars Sondergaard, Derk Frank, Hatim Seoudy, Baland Mohammad, Oliver J. Müller, Stephanie Sellers, David Meier, Janarthanan Sathananthan, Georg Lutter

**Affiliations:** 1Department of Cardiovascular Surgery, University Medical Center Schleswig-Holstein, Campus Kiel, 24105 Kiel, Germany; zhang_xiling@outlook.com (Z.X.); thomas.puehler@uksh.de (T.P.); baland.mohammad@uksh.de (B.M.); 2DZHK (German Centre for Cardiovascular Research), Partner Site Hamburg/Kiel/Lübeck, 24105 Kiel, Germany; derk.frank@uksh.de (D.F.); oliver.mueller@uksh.de (O.J.M.); 3Rigshospitalet, Copenhagen University Hospital, 2100 Copenhagen, Denmark; lars.soendergaard.01@regionh.dk; 4Department of Internal Medicine III (Cardiology, Angiology, and Critical Care), University Medical Center Schleswig-Holstein, Campus Kiel, 24105 Kiel, Germany; hatim.seoudy@uksh.de; 5Centre for Cardiovascular Innovation, St Paul’s and Vancouver General Hospital, Vancouver, BC V6Z 1Y6, Canada; ssellers@providencehealth.bc.ca (S.S.); dmeier@providencehealth.bc.ca (D.M.); jsathananthan@providencehealth.bc.ca (J.S.); 6Cardiovascular Translational Laboratory, St Paul’s Hospital & Centre for Heart Lung Innovation, Vancouver, BC V6Z 1Y6, Canada; 7Centre for Heart Valve Innovation, St. Paul’s Hospital, University of British Columbia, Vancouver, BC V6Z 1Y6, Canada

**Keywords:** mitral regurgitation, mitral insufficiency, transcatheter mitral valve repair, transcatheter mitral valve replacement, transcatheter technology, percutaneous, transluminal, endovascular, access, approach

## Abstract

Over the last two decades, transcatheter devices have been developed to repair or replace diseased mitral valves (**MV**). Transcatheter mitral valve repair (**TMVr**) devices have been proven to be efficient and safe, but many anatomical structures are not compatible with these technologies. The most significant advantage of transcatheter mitral valve replacement (**TMVR**) over transcatheter repair is the greater and more reliable reduction in mitral regurgitation. However, there are also potential disadvantages. This review introduces the newest TMVr and TMVR devices and presents clinical trial data to identify current challenges and directions for future research.

## 1. Introduction

Mitral valve (MV) disease is the most common heart valve disease, with a prevalence in western countries of 1% to 2% in the general population and a prevalence of 10% in persons over 75 years of age [1]. In the last decades, rheumatic heart diseases have decreased dramatically in developed countries but, due to an aging population, the incidence of mitral regurgitation (MR) has gradually surpassed that of aortic valve stenosis, ranking first in valvular disease [1,2].

MR is a disease in which the MV does not close adequately during left ventricular systole, resulting in regurgitation of blood from the left ventricle (LV) to the left atrium, and includes primary (degenerative) MR and secondary (functional) MR [3]. Primary MR is mainly due to degenerative MV disease resulting in anatomical changes in the valve leaflets and chordal that cause MR; the recommended treatment for severe primary MR is surgery. Secondary MR is mainly due to ischemic or non-ischemic left ventricular failure with an enlarged mitral annulus, or dilatation of the left atrium in atrial fibrillation.

Optimization of pharmacological therapy is the first step in treating all patients with secondary MR, and the application of cardiac resynchronization therapy requires a comprehensive evaluation according to the relevant guidelines [4]. The European Society of Cardiology/European Association for Cardio-Thoracic Surgery guidelines recommend either surgery (class IIa) or catheter intervention (class IIb) for patients with secondary MR who have persistent symptoms despite conventional optimal heart failure therapy [4].

In elderly patients and patients with comorbidities, the surgical risk is high and approximately 50% of patients with severe MR symptoms are not suitable candidates for open-heart surgery [5]. The morbidity and mortality rates during hospitalization after MV repair and MV replacement in patients aged 80 to 89 years have been reported to be 6% and 13%, respectively [6]. Therefore, for elderly MR patients with comorbidities, there is an urgent need for an appropriate, less invasive treatment. The development of transcatheter mitral valve therapy offers new options for high-risk patients with MR. Many of these patients have benefited from transcatheter mitral valve repair **(TMVr)**. However, there are still patients who are anatomically unsuitable for these therapies, such as patients with a high coaptation defect or severe mitral valve calcification. As a result, interest in transcatheter mitral valve replacement (TMVR) has increased over the last few years.

This review covers **TMVr** and **TMVR** devices, early results of treatment, challenges to treatment, and scientific views on the future direction in this constantly evolving field.

## 2. Transcatheter Mitral Valve Repair (TMVr)

The different components of the mitral valve (leaflets, annulus, chordae, papillary muscles, and LV) and the different pathogeneses of the disease (primary and secondary) have led to a series of different therapeutic measures, such as transcatheter edge-to-edge repair (TEER), direct/indirect annuloplasty, and chordal repair. An overview of the features of transcatheter, mainly transfemoral mitral valve repair devices that have received CE make approval is indicated in Table 1. Table 2 shows the clinical trials currently being conducted.

### 2.1. TEER Devices

MitraClip^TM^ (Figure 1A)

The MitraClip^TM^ device (Abbott Laboratories, North Chicago, IL, USA) is based on the traditional surgical “edge-to-edge” procedure and consists of two main components: a clip and a catheter system. After transseptal puncture, and under the guidance of transesophageal echocardiography (TEE) and fluoroscopy, the clip is advanced in the left atrium through the catheter. The anterior posterior mitral leaflets are grasped together, according to the anatomical location of the regurgitant jet, to create a double-orifice outflow tract. The latest fourth-generation MitraClip (MitraClip G4) offers more clip sizes for tailored repair. It also has a new leaflet grasping technology called a Controlled Gripper Actyation^TM^, which allows physicians to grasp leaflets simultaneously or independently to confirm and optimize leaflet insertion. The MitraClip G4 offers four types of clips based on mitral valve anatomy: NTW, NT, XTW and XT. The recommendations of clip selection are shown in Table 3. The EVEREST II study—the first published RCT on MItraClip—included 279 patients with severe (mainly primary) MR who were randomized in a 2:1 ratio to the MitraClip group (n = 184) and the surgical group (n = 95) [7]. The results showed that the incidence of adverse events was significantly lower in the MitraClip group than in the surgical group (15% vs. 48%, *p* = 0.001). A 5-year follow-up revealed no significant difference in mortality between the MitraClip and surgical groups (20.8% vs. 26.8%, *p* = 0.4), nor was there a significant difference in New York Heart Association (NYHA) class between the two groups [25]. More recently, the COAPT [10] and MITRA-FR [26] RCT, which enrolled secondary MR patients, evaluated treatment with MitraClip plus guideline derived medical therapy versus the guideline derived medical therapy alone. The COAPT trial showed a reduction in long-term mortality and rehospitalization rates for heart failure at 2 years in the MitraClip group. The MITRA-FR trial found that people treated with MitraClip and those treated with medical management had similar rates of rehospitalization for heart failure and comparable mortality rates. These discordant results are at least in part related to different inclusion criteria in the two trials and highlighted the importance of patient selection in order to maximize the benefit of treatment with MitraClip. Indeed, secondary MR is a disease process within the LV and the typical MR classification ignores the importance of LV. Grayburn et al. [27] found that patients in the COAPT trial had a higher effective regurgitant orifice area with a lower left ventricular end-diastolic volume (LVEDV) (disproportionate MR). In contrast, in the MITRA-FR trial, patients had MR proportional to the degree of LV dilatation (proportionate MR). Thus, the characteristics of MR proportional or not to LVEDV appears to be critical for correctly selecting patients susceptible of deriving optimal benefits from MitraClip. Accordingly, Pibarot et al. [28] suggested that MitraClip may not be suitable for patients with secondary MR in the context of LVEF, 20% and LV end diastolic diameter > 70 mm. Michael J Mack et al. [29] reported the 3-year follow-up of the COAPT trial, confirming initial positive results. Thus, the annualized rates of heart failure hospitalizations per patient-year were 35.5% with TMVr and 68.8% with guideline derived medical therapy alone. Patients who received TMVr also sustained improvements in MR severity, quality-of-life measures, and functional capacity for 3 years. Interestingly, 58 patients treated with guideline derived medical therapy alone, crossed over to TMVr, had a reduction in subsequent composite rate of mortality or heart failure hospitalization compared with those who continued on guideline derived medical therapy alone.

As TEER procedure becomes more popular, the number of patients with residual or recurrent MR is rapidly increasing [30,31]. In published data, the failure rate of MitraClip procedures ranged from 4.8% to 9.5%, and the recurrent MR rate was 5.1–21.5% [32,33,34]. EI-Shurafa et al. [35] proposed that surgical intervention could be used to improve survival in patients with residual MR or recurrent MR after MitraClip procedure.

Overall, the MitraClip has been shown to have a high safety and efficacy profile in adequately selected patients, with a low incidence of complications (atrial septal defect, bleeding, pericardial effusion, endocarditis, clip detachment, clip embolization, mitral stenosis, and other device-related complications). Future studies will need to answer questions regarding patient selection, long-term outcomes and the importance of residual MR following MitraClip and post-procedural transmitral pressure gradients [36], which have recently been questioned.

PASCAL (Figure 1B)

The PASCAL system (Edwards Lifesciences, Irvine, CA, USA) is another form of TEER [39] In contrast to the first three generations of MitraClip, the PASCAL “clamp” is broader, and each paddle can be activated separately, allowing for independent leaflet capture. Moreover, there is a large central spacer to fill the regurgitant orifice. In addition, the device can be extended to facilitate manipulation in the LV [40]. In contrast to the MitraClip system, the implant closure does not require activation of the locking element, but rather the implant is passively maintained closed by the nitinol shape-memory. In addition, the PASCAL delivery system provides continuous left atrial pressure monitoring

The first clinical trial of the PASCAL system was applied to 23 patients with severe MR who were inappropriate for surgery [41], of which 22 patients (96%) had a postoperative residual MR volume less than grade 2. During the 30-day follow-up, three patients (13%) died, and 19 of the 20 patients (95%) who survived were in NYHA class I or II. In addition, PASCAL has been shown to be a safe and effective treatment for severe primary or secondary MR in 109 patients in the CLASP trial [14] as well as in a smaller study by Kriechbaum et al. [42] Two-year results from the CLASP trial showed that MR ≤ 1+ was achieved in 78% of patients and MR ≤ 2+ was achieved in 97% of patients. Of the patients, 93% were in NYHA class I to II [43]. Early outcomes from the CLASP IID trial demonstrated that MR ≤ 1+ was achieved in 73% and ≤2+ in 98% of patients, with 89% of patients in NYHA class I/II during the 30-day follow-up [44].

### 2.2. Annuloplasty Devices

Cardioband (Figure 1C)

The Cardioband system (Edwards Lifesciences, Irvine, CA, USA) secures the annuloplasty band to the posterior annulus through small anchors under TEE and X-ray guidance, and then adjusts the annuloplasty band to reduce the diameter of the mitral annulus. In a multicenter clinical study [16], the Cardioband system was used for the treatment of 60 patients with moderate to severe MR. One year later, the survival rate was 87%, with 61% of patients with mild or less MR, and patients showed significant improvements in heart function, quality of life, and exercise capacity. However, the Cardioband system lacks sufficient evidence-based medical evidence, and more clinical trials are needed to further validate its safety and efficacy.

Mitralign (Figure 1D)

The Mitralign system (Mitralign Inc., Tewksbury, MA, USA) is introduced from the femoral artery, retrograde through the aorta into the LV and left atrium, and paired surgical pledgets are anchored across the annulus. The pledgets are pulled together to reduce the diameter of the annulus. Nickenig et al. [18] reported 71 patients with moderate to severe functional MR treated with Mitralign, and 50 (70.4%) were successfully implanted with no mortality. The all-cause mortality rates at 1 and 3 months were 4.4% and 12.2%, respectively. Echocardiography showed MR reduction in 50% of treated patients by a mean of 1.3 grades, at 6 months.

Carillon (Figure 1E)

The Carillon mitral contour system (Cardiac Dimensions, Washington, DC, USA) is the only device in its category, allowing indirect annuloplasty without the need for transseptal puncture. The procedure is guided by X-ray to reach the right atrium via the internal jugular vein, and the device is deployed after entering the coronary sinus. The diameter of the mitral annulus can be shortened by shortening the length of the device after insertion. In a randomized clinical trial of REDUCE-FMR [19], 120 patients were divided into a Carillon treatment group (87) and a control group (33 treated with drugs). The results showed a significant MR reduction and a significant reversal of ventricular remodeling in the Carillon group compared to the control group. However, there are several limitations of the Carillon mitral contour system that hinder the development of this device. First, the position of the coronary sinus is not necessarily coplanar with the mitral annulus, and second, placement of the device may lead to serious complications such as compression of the coronary arteries and damage to the cardiac conduction system, cannot be used in patients with pacemaker-lead in the coronary sinus for cardiac resynchronization therapy.

### 2.3. Chordal Repair

NeoChord (Figure 1F)

The NeoChord system (NeoChord Inc., St. Louis Park, MN, USA) is a device used to treat primary MR caused by MV prolapse/flail posterior. In contrast to the above-mentioned TMVr devices, this device is guided by TEE through an apical approach into the LV, with one end connected to the MV leaflet and the other to the left ventricular myocardium, forming an artificial chord fixed to the ventricular wall. A Trans-Apical Chordae Tendineae trial demonstrated promising immediate safety and efficacy of the NeoChord system with achieved acute procedural success (placement of at least one neo-chord and reduction of MR from 3+ or 4+ to ≤2+) in 26 patients [45]. A clinical trial reported 213 patients treated with this device and 206 (96.7%) had a successful procedure. One year later, the morbidity and mortality rates were 1.9% and 7.9% of patients with severe MR. This study demonstrated the safety, efficacy, and reproducibility of the NeoChord system [23]. The ongoing RECHORD trial (NCT02803957) is comparing the NeoChord system with open surgical mitral valve repair in degenerative MR. Furthermore, one compassionate-use case already received a successful NeoChord implantation by a transfemoral approach by the Mainz group. 

## 3. Transcatheter Mitral Valve Replacement (TMVR)

MV disease is complex as well as heterogeneous, and TMVr devices are difficult to fully address all variabilities in MV anatomy and patients’ conditions. The development of TMVR offers a new treatment option to address MR. TMVR and may have several theoretical advantages over TMVr, namely predictably reducing MR, and possibly being less invasive than surgical procedures [46]. The initial TMVR clinical experience involved the following three main conditions: (1) a valve-in-valve procedure for patients with MV bioprosthesis degeneration [47,48]; (2) a valve-in-ring procedure for patients with annuloplasty rings [49,50], and (3) a valve-in-native ring procedure for patients with severe calcification of the mitral annulus [51,52]. In the case of a surgical bioprosthetic valve, some cases of annuloplasty rings, and some calcified native mitral annulus, the annular morphology offers enough support and stability to accomplish TMVR with existing valves for transcatheter aortic valve replacement (TAVR) (i.e., the Sapien valve). 

Indeed, surgery is still the standard approach to MR treatment, and the transcatheter options for repeat procedures in patient with previous mitral surgery is highly relevant, as these patients are often at too high-risk for repeat surgery. To date, the current literature reports mitigated results and significant morbidity in some of these situations. Thus, the VIVID registry [53] reported that hemodynamics after valve-in-valve and valve-in-ring procedures were suboptimal. In particular, the 4-year mortality rate after the valve-in-ring procedure was almost 50%. The TVT registry [54] showed a 22.3% mortality rate at 1-year after valve-in-valve procedure in patients with an STS score > 8. For valve-in-mitral annular calcification (MAC) patients, the study showed that all-cause 30-day mortality was 34.5%, and 1-year all-cause mortality was 62.8% [55,56]. Strategies must thus be developed to optimize procedural results in this challenging clinical setting.

Nevertheless, since most of MR patients do not have previous surgery or significant calcification of their mitral annulus, the valved stents used for TAVR cannot be used for TMVR.

The valve-in-native valve procedure for these patients is genuine TMVR. Over 30 TMVR devices are currently in development, and the field is in constant expansion [57,58]. Here, we focus on the devices currently in clinical evaluation. Table 4 shows an overview of these devices.

### TMVR Devices

Tendyne Mitral Valve System (Abbott Laboratories, IL, USA) (Figure 2A)

The Tendyne mitral valve is the only TMVR device with a CE mark (since January 2020). The Tendyne mitral valve system is a self-expanding tri-leaflet porcine pericardial valve mounted on a nitinol frame, which is fully repositionable and retrievable. 

Its design has many advantages as follows: (1) the D-shaped design prevents left ventricular outflow tract obstruction (LVOTO); (2) it can be retrieved and re-released or adjusted when the implantation position or the efficacy is unsatisfactory; (3) the presence of an atrial cuff prevents perivalvular leakage, and (4) the reliance on the apical tether rather than clamping of leaflets or chordae is the most unusual feature of the Tendyne valve and the most unique in its design. The apical tether provides strong tensile force, virtually eliminating the risk of atrial embolization of the valve; secondly, there is no need to clamp the leaflets or chordae by using the apical tether because the stent on the ventricular portion can be narrowed towards the center. By adjusting the position of the tether, the valved stent can be drawn toward the free wall of the ventricle, mitigating the risk of LVOTO. The apical pad can also serve to seal the myocardial orifice created with transapical puncture. Thirteen sizes of this Tendyne valved stent are available. 

The first in-human implantation of the Tendyne valve was performed in February 2013 and was reported as a two-patient series the following year. A dramatic improvement in intracardiac pressures, along with complete elimination of MR was reported [66]. 

In the Tendyne global feasibility trial, one-hundred patients were enrolled in multiple centers from November 2014 to November 2017 (mean age 75.4 ± 8.1 years, secondary MR n = 89, primary MR n = 11). This prospective non-randomized study evaluated 30-day and 1-year outcomes following transapical TMVR with the Tendyne prosthesis [59]. 

The results demonstrated technical success in 97% of patients, and no perioperative mortality. At 30 days, 98.8% of patients presented with no or trace regurgitation. The all-cause mortality was 6% after one-month. Furthermore, the all-cause mortality was 26% with no MR in 98.4% at 1-year. A small study showed encouraging results of the Tendyne system in patients with severe MAC, for which treatment options are currently limited. The device was successfully implanted with correction of MR in nine patients, and there were no procedural deaths. One patient presented with LVOTO (valve malrotation) and required alcohol septal ablation. There was one cardiac death and one non-cardiac death in the follow-up (median 12 months). Clinical improvement with mild or no symptoms occurred in all patients alive at the end of follow-up [67].

The SUMMIT trial (NCT03433274) is an ongoing prospective, controlled, multicenter clinical investigation with three trial cohorts: Randomized (Tendyne vs. MitraClip, 1:1 ratio), non-randomized, and MAC, designed to evaluate the safety and effectiveness of using the Tendyne mitral valve system for the treatment of symptomatic MR. This study should offer a large dataset regarding efficacy and safety of the Tendyne system. 

To date, 1000 Tendyne devices have been successfully implanted worldwide.

Tiara TMVR System (Neovasc Inc., Richmond, BC, Canada) (Figure 2B)

The Tiara TMVR system has a self-expanding nitinol frame with three bovine pericardial leaflets. The device is D-shaped and fits geo-magnetically in the native mitral annulus. The valve features three anchors (two anterior and one posterior) on the ventricular part [68]. The ventricular anchors are designed to secure the valve (the fibrous trigone anteriorly and posterior shelf of MV annulus) which may prevent migration and reduce the risk of paravalvular leakage, LVOTO, as well as coronary ostial encroachment [68]. The valve is implanted transapically and comes in two sizes (35 mm: internal dimensions 30 × 35 mm, area 6.3–9 cm^2^; 40 mm: internal dimensions 34.2 × 40 mm, area 9–12 cm^2^) [69].

The first in-human implantation was reported in January 2014 [69]. The two major Tiara TMVR system trials, TIARA I (Early Feasibility Study of the Neovasc Tiara Mitral Valve System) (NCT02276547) and TIARA II (Tiara Transcatheter Mitral Valve Replacement Study) (NCT03039855), are ongoing and showed promising preliminary results in 71 patients with a 94% technical success rate and a 30-day mortality rate of 11.3 [60,61].

Intrepid TMVR System (Medtronic, Minneapolis, MN, USA) (Figure 2C)

The Intrepid TMVR system integrates a self-expanding nitinol frame with tri-leaflet bovine pericardial valve, which includes an inner stent with valve attached and an independent conformable outer stent to engage the annulus and leaflets, providing fixation while isolating the inner stent from the dynamic anatomy [70]. The outer stent includes a flexible brim designed to aid echocardiography imaging. 

Bapat et al. [62] described the implantation of the Intrepid TMVR system in the first 50 patients with a 30-day follow-up. One patient had a complication of apical hemorrhage and implantation was discontinued, while 48 of the remaining 49 patients were successfully implanted. Mortality rate at 30 days was 14%, with none to mild MR in all surviving patients. The Apollo trial (NCT03242642) began in 2017 and is expected to enroll 1350 patients. The primary endpoint is a composite of 1-year all-cause mortality, stroke, reoperation (or reintervention), and cardiovascular hospitalization rates, with estimated primary completion in October 2023 and estimated study completion in October 2028. The CE approval has not yet been granted.

EVOQUE TMVR System (Edwards Lifesciences, Irvine, CA, USA) (Figure 2D)

The EVOQUE (Edwards Lifesciences, Irvine, CA, USA) valve is a transseptal self-expanding nitinol valve with bovine pericardial leaflets. The atrial part provides additional annular anchorage and contains a paravalvular sealing skirt, which is designed to minimize paravalvular leakage. Two sizes (44 and 48 mm) are currently available and are delivered via a transfemoral/transseptal approach. The delivery system allows for three planes of motion, permitting coaxial alignment and precise positioning within the annulus. To reduce the risk of LVOTO, the delivery system allows the valve to be tilted before deployment. An early feasibility trial is currently enrolling (NCT02718001). The results of the first 14 patients treated with the EVOQUE valve showed technical success in 93% of patients and one patient undergoing surgical conversion. Two patients underwent paravalvular leak closure, and one patient underwent alcohol septal ablation for LVOTO. Of the patients, 93% survived at 30-days. MR was eliminated in 80% of patients, and the remaining 20% of patients had mild MR [63].

SAPIEN M3 System (Edwards Lifesciences, Irvine, CA, USA) (Figure 2E)

The SAPIEN M3 system is a modification of the SAPIEN 3 TAVR system, including a nitinol dock with a balloon-expandable tri-leaflet bovine pericardial valve. The SAPIEN M3 valve adds a polyethylene terephthalate (PET) skirt to minimize paravalvular leakage. Early experience in 10 patients showed promising safety and efficacy, with nine successfully implanted patients with no significant adverse events [71]. Results from a recent early feasibility study (NCT03230747) demonstrated technical success in 89% of 35 patients. All-cause mortality rate was 2.9% (n = 1), with one disabling stroke at 30 days. Echocardiographic data were available for 33 of 34 patients; 88% of patients had MR ≤ 1+ [64]. The ENCIRCLE will study the safety and efficacy of the SAPIEN M3 system in 400 patients and recently started patient recruitment (NCT04153292). The estimated primary completion date is February 2024, and the estimated study completion date is February 2028.

HighLife TMVR system (HighLife Medical, Paris, France) (Figure 2F)

The HighLife TMVR system’s special component is a sub-annular implant ring that acts as a docking system. A transfemoral retrograde transaortic approach is used to place a sub-annular ring around the MV from the start to act as an anchor for the self-expanding tri-leaflet bovine pericardial valve. This design could theoretically reduce the risk of perivalvular leakage and LVOTO. The first two case of HighLife implantation in humans showed excellent early hemodynamic performance [72]. Data from the first 15 patients showed that 13 patients were successfully implanted, and two of them (13%) were switched to surgery. Thirty-day-mortality was 20%, and LVOTO occurred in one patient. There was no mild or greater MR in the successful implantations [65].

In addition to the systems mentioned above, other technologies are under development and are still in their early stages, with only a few cases being reported. Other devices under development include the NAVI System (NaviGate Cardiac Structures Inc., Lake Forest, USA); the AltaValve TMVR system (4C Medical Technologies, Inc., Maple Grove, MN, USA); the Cephea TMVR System (Cephea Valve Technologies, Abbott Inc., San Jose, CA USA) (Table 5).

## 4. Current Challenges and Discussion

Different strategies should be adopted for MR with different etiologies (Figure 3). The TEER has the most research data and the clearest evidence of therapeutic efficacy for all causes of MR. Annuloplasty can only be used in patients with secondary MR, but more excellent development may lie in the future in conjunction with leaflet repair. Chordal repair is safe but has relatively limited indications, is more effective in central posterior leaflet (P2) prolapse and has only been studied in low-risk patients; data are still needed to support safety, efficacy, and long-term outcomes in high-risk patients. Atrial functional MR is a specific type of secondary MR with a unique pathophysiology that includes isolated annular dilation [74] and insufficient compensatory leaflet growth [75]. Prevention of left atrial dilation and restoration of sinus rhythm may be the key to treating atrial functional MR. Prospective trials comparing rhythm recovery with surgical/endovascular strategies are to be expected [76,77,78]. However, despite the minimally-invasive transfemoral approach, low mortality rate, and quick recovery after TMVr, some disadvantages inherent to this approach and to the complexity of MV disease and anatomy are undeniable. The fundamental disadvantage of TMVr is that MR reduction is less predictable, and MR might persist or re-occur. In patients with functional MR, recurrence rates may be greater due to further cardiac remodeling and the repair device, which does not fully occlude the MV in systole [79]. In addition, transcatheter repair procedures are often technically challenging and, in certain cases, a combination of devices is required to ensure the procedure’s effectiveness. The use of TEER in combination with direct annuloplasty or tendon cord repair has been reported to achieve complementary results [80,81,82,83]. However, the optimal sequence of combined techniques is still undetermined and needs to be further studied in randomized clinical trials.

Notably, the role of center and operator experience cannot be ignored. Reports have already illustrated that increased institutional and operator experience is associated with improvements in procedure success, procedure time, and procedural complications for TMVr with MitralClip [84,85].

TMVR has the potential to overcome these limitations even if some challenges remain to be solved. Based on experience with TAVR, the transfemoral approach would be the preferred interventional approach. However, due to the larger size of the MV compared to the aortic valve, TMVR devices require a large profile delivery system, and most devices still require a standard transapical approach. Thus, it is predictable that perioperative complications might be further reduced when transseptal TMVR delivery becomes more widely feasible. Therefore, the development of maneuverable low-profile delivery systems should be key in future research. This might allow reduction in the need for atrial septal defect closure following transseptal approach.

TMVR offers some additional challenges. Indeed, compared to TAVR, the TMVR faces more problems and challenges due to the more complex anatomy of the MV apparatus: (1)The mitral annulus is saddle-shaped and D-shaped, not circular, and is not in the same plane. Even if a skirt is placed on the atrial portion, paravalvular leakage may still occur.(2)In TAVR, the aortic valve is calcified, rigid, and rounder after pre-dilation, which makes it relatively easy to anchor the circular aortic valved stent to the native annulus in a tube-like area. In contrast, the mitral annulus is compliant, and its shape is constantly changing with the cardiac cycle and underlying pathological process. Thus, it generally does not provide radial support for the new valved stent because the annulus is located between the contracting left atrial and ventricular chamber. Thus, mitral fixation has to be done in a very different way than that for TAVR and engineers face significant challenges when developing new devices.(3)The intracavitary pressure from the LV contraction can be high (180 mmHg), and the prosthetic valve is at risk of atrial embolization.(4)There are about 24 chordae in the left ventricular cavity which can interfere with the implantation and fixation of the new prosthetic valve.(5)The probability of acute LVOTO after TMVR is 8.2% [86], rising to 9.2% if there is calcification of the mitral annulus. In the case of valve in MAC, the 30-day LVOTO rate was 39.7%, and the all-cause mortality reached to 34.5% [55]. LVOTO is related to a variety of factors, including mainly the angle between aortic and mitral annulus, degree of atrial septal hypertrophy, length of anterior mitral leaflet, and the size of the LV [87]. Nevertheless, LVOTO can be predicted by preoperative cardiac 3D-computed tomography (CT). 

Some approaches have been proposed to overcome these challenges: the use of a D-shaped valve ring, the alcohol septal ablation method, the anterior leaflet laceration technique (Lampoon) that can minimize the impact on left ventricular outflow tract [88], the fixation of a mitral valved stent by clamping valve leaflets or chordae, and the use of an atrial skirt design or even the use of a neo-chord that does minimize paravalvular leakage (e.g., Tendyne valve).

Although current mitral valved stents meet basic requirements, such as fixation to the mitral annulus, satisfactory valve function, no paravalvular leakages, and no short-term complications, they still face some practical problems, especially when used in patients with lower surgical risk (lower Society of Thoracic Surgeons, STS score) or at a younger age when optimal long-term results are mandatory. 

These issues include:(1)In the vicinity of the mitral valved stent, where blood flow is maintained at a very low velocity within a relatively small circulatory area, the potential for blood clotting in the left atrium is increased. Indeed, to prevent paravalvular leakage, prosthetic valves are designed to have an atrial skirt and a complex structure aligned towards the atrial portion, where blood flow is slow and therefore prone to thrombus formation. In addition, the peripheral area of the mitral valved stent against the ventricular wall is also blinded to blood flow and is also susceptible to thrombus formation. Clinical studies have shown that thrombosis is, indeed, a problem and a major cause of postoperative death in many patients. Some studies suggest adequate oral anticoagulants as one of the main solutions [89,90,91,92]. Valve leaflet thrombosis has been seen in early TMVR systems, but the optimal antithrombotic strategy has not yet been determined. In the early Tendyne experience, 6% of patients presented thrombus, resulting in patients having to be anticoagulated with warfarin for more than 3 months [59]. With the EVOQUE and SAPIEN M3 systems, all patients underwent anticoagulation after implantation. Further research is needed regarding the optimal duration of anticoagulation and the dose of anticoagulant drugs [71,93]. This is a critical consideration compared to transcatheter repair, which does not require anticoagulation in patients with sinus rhythm. Four-dimensional multilayer spiral CT has a high predictive value for postoperative device thrombosis and may be routinely used after TMVR [94].(2)The mitral annulus size may easily be underestimated. Nakashima et al. [95] reported modest changes in mitral annulus geometry (7.2–13.9%), resulting in size alignment changes (24.2%) in a significant proportion of patients with the Intrepid TMVR, suggesting that size is essential for TMVR devices. The optimal TMVR valve must balance the need to accommodate a large mitral annulus while minimizing LVOT interactions. Moreover, larger TMVR devices that can accommodate larger mitral annulus come at the cost of a high-profile delivery system, limiting transseptal delivery [96].(3)The durability of prosthetic valves may also be an issue. The prosthetic aortic valve is implanted in the aortic root where there is little local tissue activity, and therefore valve degeneration is low after TAVR (about 6.6% after 5 years [97]). In contrast, the mitral annulus, chordae and papillary muscles undergo contractile motion in response to the cardiac cycle. The mechanical damage sustained over time can be staggering. Interestingly, in the case of the Tendyne valve, the apical tether is mechanically strained during each cardiac cycle, but this does not appear to be problematic up to seven years after TMVR [59,66,98]. An analogy is the Melody pulmonary valve, which is implanted in the right ventricular outflow tract, which is contractile, resulting in a 1-year fracture rate of 15% after Melody pulmonary valve implantation, with fracture rates of 25% at 2 years after implantation [99]. 

The current medical consensus is that mitral valves should not be replaced if they can be repaired because MV replacement may damage the chordae tendineae, which are part of the ventricular systolic function. When the ventricle contracts, the chordae pull the mitral apparatus to prevent prolapse and regurgitation, as well as pull the apical tissue toward the MV to help the ventricle contract. Long-term lack of chordae inevitably leads to impaired ventricular function, which in severe cases contributes to heart-failure worsening. Therefore, there is currently a preference for MV replacement surgery that preserves the sub-valvular apparatus, such as chordae tendineae [100]. All TMVR techniques fully maintain the valvular apparatus including the chordae independently of their access. The current TMVR delivery approaches are illustrated in Figure 4. There are two main approaches for TMVR delivery: transseptal and transapical access. The transseptal approach is the less invasive but is technically challenging because of its small operating space and its large atrial septal defect. Currently, the transapical approach is most frequently used, since it is faster and allows for precise and perpendicular positioning, anchoring, and placement of the prosthetic valve. However, a series of adverse effects such as bleeding and myocardial injury may occur.

Whether with TMVr or TMVR, patients need accurate quantification of mitral regurgitation, assessment of mitral valve anatomy, assessment of LV dimensions and systolic function, as well as left atrial dimensions. In addition, the mitral valve cannot be inspected during the intervention procedure compared with surgery, so it is more important to accurately visualize the mitral valve apparatus. Three-dimensional TEE has been shown to provide better information on valve morphology and function with assessing the mitral annulus perimeter and area [102,103,104]. In addition, three-dimensional TEE showed excellent agreement with cardiovascular magnetic resonance (CMR) in quantifying effective regurgitant orifice area and regurgitant volume [105,106,107,108]. It also plays an important role in procedural guidance. Multidetector row computer tomography (MDCT) is also an essential assessment technique. MDCT is the imaging technique of selecting patients and planning transcatheter treatment for mitral annuloplasty or valve replacement. MDCT is also useful for planning transseptal and transapical implantation routes [109]. It is critical to predict LVOTO by calculating the neo left ventricular outflow tract (LVOT) area when considering TMVR. If the neo LVOT area is measured throughout the entire cardiac cycle (multiphase average), or if the neo LVOT area is measured at early systole, potentially suitable patients will probably be identified [110,111]. CMR is also a guideline-recommended technique for the evaluation of MR [4,112]. In patients with primary MR, CMR can be used to assess the effect of MR on LV dimensions and function [113]. In patients with secondary MR, the use of CMR to assess the extent of myocardial fibrosis/scar is of prognostic importance [114].

## 5. Conclusions

Overall, TMVR has the theoretical advantages of being applicable to more patients and could achieve “one valve for the individual patient.” It also potentially offers a more complete and predictable MR reduction, and might be technically less challenging than TMVr. Nevertheless, TMVR comes with a potentially higher burden of complication, and long-term follow-up data are still lacking. TMVr, on the other hand, can only fit a limited number of patients and can be only performed by highly-experienced operators but has a little detrimental physiologic impact on the valve because it leaves most of its anatomical structures untouched. In the near future, it will be interesting to observe the progress of the SUMMIT trial as it compares the outcomes of the Tendyne system with those of the MitraClip system.

Thus, it appears that for now, repair and replacement remain complementary rather than competitive.

## Figures and Tables

**Figure 1 jcm-11-03377-f001:**
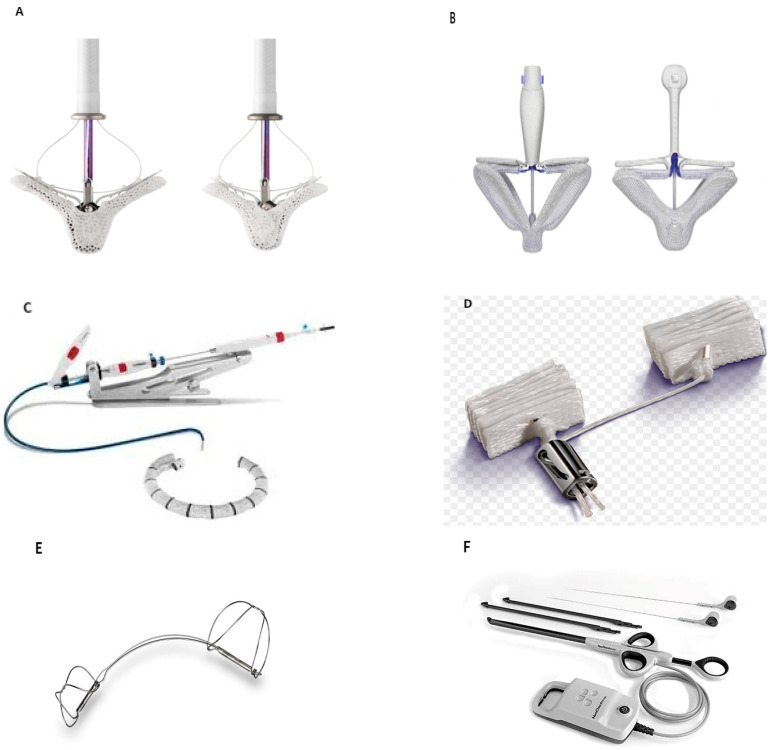
Transcatheter Mitral Valve Repair Systems: (**A**) MitraClipTM device. Image courtesy of Abbott. (**B**) PASCAL system. Image courtesy of Edwards Lifesciences. (**C**) Cardioband system. Image courtesy of Edwards Lifesciences. (**D**) Mitralign system. Image courtesy of Mitralign Inc. (**E**) Carillon mitral contour system. Image courtesy of Cardiac Dimensions. (**F**) NeoChord system. Image courtesy of NeoChord Inc.

**Figure 2 jcm-11-03377-f002:**
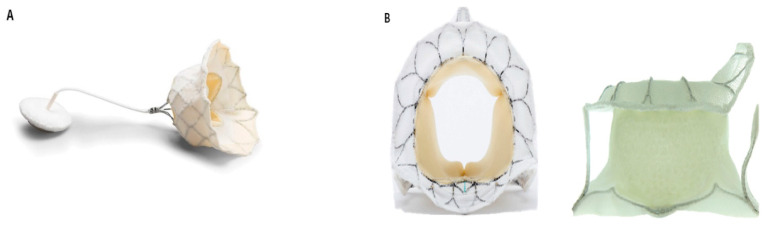
Transcatheter Mitral Valve Replacement Systems: (**A**) Tendyne prosthesis. Image courtesy of Abbott. (**B**) Tiara prosthesis (left: top view, right: side view). Image courtesy of Neovasc Inc. (**C**) Intrepid prosthesis. Image courtesy of Medtronic. (**D**) EVOQUE prosthesis. Image courtesy of Edwards Lifesciences. (**E**) SAPIEN M3 prosthesis. Image courtesy of Edwards Lifesciences. (**F**) HighLife prosthesis. Image courtesy of HighLife Medical.

**Figure 3 jcm-11-03377-f003:**
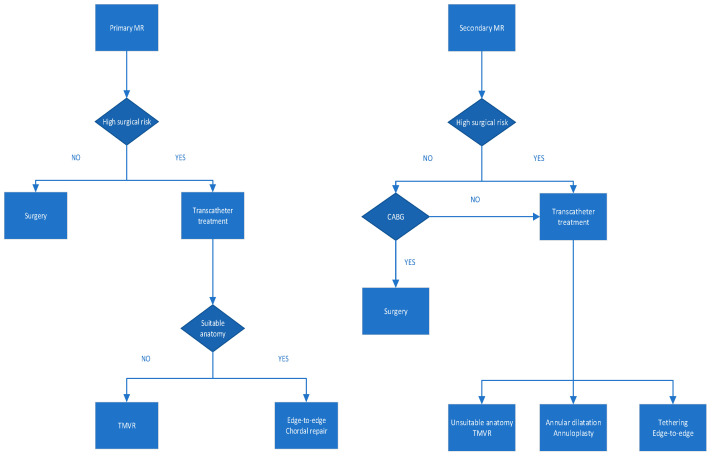
Selection of the most appropriate therapeutic strategy. CABG: coronary artery bypass grafting.

**Figure 4 jcm-11-03377-f004:**
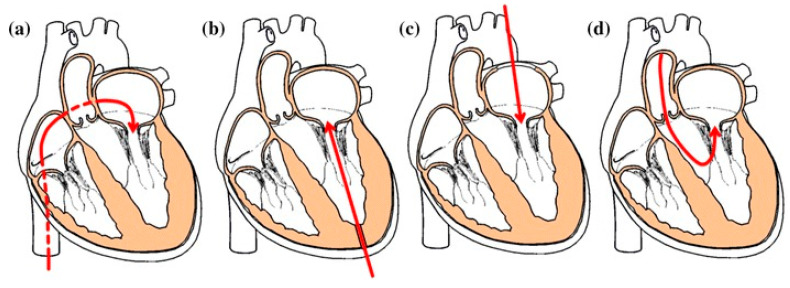
Transcatheter mitral valve delivery approaches: (**a**) transseptal, (**b**) transapical, (**c**) left atriotomy, (**d**) transaortic [101].

**Table 1 jcm-11-03377-t001:** Overview of Transcatheter Mitral Valve Repair Device Features.

Device	Repair Method	Approach	Indications	30-Day Mortality Rate
MitraClip^TM^	TEER	transseptal	Primary/Secondary MR	0.9–6% [7,8,9,10,11,12,13]
PASCAL	TEER	transseptal	Primary/Secondary MR	1.6–2% [14,15]
Cardioband	Direct annuloplasty	transseptal	Secondary MR	3.3–5% [16,17]
Mitralign	Direct annuloplasty	transseptal	Secondary MR	4.4% [18]
Carillon	Indirect annuloplasty	transseptal	Secondary MR	1.9–2.7% [19,20,21,22]
NeoChord *	chordal repair	transapical/transeptal	Primary MR	0–1.9% [23,24]

* Neochord is the only device which is mainly implanted transapically. TEER, transcatheter edge-to-edge repair.

**Table 2 jcm-11-03377-t002:** Ongoing Trial of Transcatheter Mitral Valve Repair Device.

Trial	Device	Aim
MITRA-HR RESHAPE-HF2 MATTERHORN REPAIR-MR	MitraClip	Long-term outcomes Risk stratification Patient selection
CLASP IID/IIF	PASCAL	Safety and effectiveness compared with MitraClip
MiBAND ACTIVE	Cardioband	Post-Market approval safety and efficacy (MiBAND) Identify optimal candidates by comparing with guideline-directed medical therapy in patients with FMR (ACTIVE)
Millipede Feasibility	Millipede	Feasibility and safety
EMPOWER	Carillon	Safety and efficacy at 5 years of follow-up
Rechord	NeoChord	Safety and effectiveness compared with open surgical repair

FMR: functional mitral regurgitation.

**Table 3 jcm-11-03377-t003:** Clip selection recommendations based on mitral valve anatomy [37,38].

Mitral Valve Anatomy	Clip Selection Recommendations
Leaflet length < 9 mm	NTW, NT
Leaflet length > 9 mm	XTW, XT
Broad jet	NTW, XTW
Smaller valve	NT
Larger valve	NTW, XTW, XT

**Table 4 jcm-11-03377-t004:** Overview of Transcatheter Mitral Valve Replacement Device Features.

Device	Anchoring Method	Approach	Indications	30-Day Mortality Rate
Tendyne Mitral Valve System	Apical tether	transapical	Secondary MR	6% [59]
Tiara TMVR System	Native leaflet engagement	transapical	Primary/Secondary MR	11.3% [60,61]
Intrepid TMVR System	Radial forces and sub-annual cleats	transapical	Secondary MR	14% [62]
EVOQUE TMVR System	External anchor	transseptal	Primary/Secondary MR	7% [63]
SAPIEN M3 System	Nitinol dock system	transseptal	Primary/Secondary MR	2.9% [64]
HighLife TMVR system	External anchor mitral annuls capture	transseptal	Secondary MR	20% [65]

**Table 5 jcm-11-03377-t005:** Features, and Studies of new TMVR Devices [73].

Device	Features	Approach	Studies
NAVI System	Nitinal self-expandable system with several annular winglets	Transaptical	No trials ongoing
AltaValve TMVR system	Self-expanding supra-annular device, with a bovine tissue valve mounted into a spherical nitinol frame	Transaptical	Early feasibility study protocol (NCT03997305), still recruiting
Cephea TMVR System	Self-expanding double-disk and trileaflet bovine pericardium tissue	Transseptal	Cephea Transseptal Mitral Valve System FIH (NCT03988946)

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
