# Peer review of "Transcatheter Mitral Valve Repair or Replacement: Competitive or Complementary?"

_jcm, 2022, doi:10.3390/jcm11123377_

Round 1
Reviewer 1 Report
This paper is an interesting review that provides an overview of current evidences of TMVr and TMVR. I have some suggestions to improve readability of the review or increasing its possibility of citation.
- I suggest to improve editing of Table 1, avoiding titles going to head. Moreover, as regards data for mortality, it is reductive to report only the data of a single study for device. I think that this Table should be improved, reporting data of mortality of principals trials and multi center international registries published since yet and reporting an extra column with ongoing trials (consult clinialtrials.gov).
- When reporting principals trials for MitraClip, I would specify that the EVEREST trial was a pioneering trial that selected mostly DMR patients with lower risk than in real-world practice. Similarly, I would specify that recent COAPT and MITRA-FR trials were conducted just in FMR patients.
- As regards future directions, I would underline also the impact of different etiology of MR upon outcome, including the new definitions of atrial functional mitral regurgitation (of which we have preliminary experience, mostly with MitraClip device). This aspect could be underlined also for other devices, for example transcatheter annular devices.
- In Discussion, the authors mainly focused upon mitral valve replacement and discussion regarding mitral valve repair is really poor. Moreover, the Discussion paragraph starts with citation of COAPT trial and the challenges regarding patients with secondary MR, who are not the main or only focus of the paper (what about DMR?). I would change the cut of discussion and I would expand this section citing the challenges that may occur with mitral valve repair, the differences (also technical) that we can meet with different etiologies and the process of choosing the different devices (in this case, are them competitive or complementary? And when or how?). In this regard, a graphical algorithm may be suitable and of good impact to the readers.
- Most of the aspects regarding TMVR have been addressed in a recent review that i suggest to cite: Luca Testa, et al. "Transcatheter mitral valve replacement in the transcatheter aortic valve replacement era." Journal of the American Heart Association 8.22 (2019): e013352.”. I would add also in this case a summary Table with all the ongoing studies regarding TMVR as reported in this paper that I suggest.
- Initial clinical experience with HighLife have been published. I suggest to add the paper “Marco Barbanti, et al. "Transcatheter mitral valve implantation using the HighLife system." JACC: Cardiovascular Interventions 10.16 (2017): 1662-1670”.
- I have a comment regarding the sentence “TMVr, on the other hand, can only fit a limited number of patients and can be only performed by highly-experienced operators”. My general opinion (but I admit that it can be absolutely personal) is that TMVR is more technically challenging that TMVr. Moreover, anatomical suitability for TMVR can be more difficult in some cases rather than TMVr. I would not be so categorical. And finally: regarding the title question (competitive or complementary?), I am not really sure that the authors answered to the question.
- Figures should be edited, as in the PDF version they resulted cut
Author Response
Dear editors and reviewers:
Thank you very much for your careful review and your very constructive suggestions with regard to our manuscript.
We have made the necessary corrections. Revised portions are marked in red in the paper. The main corrections in the paper and the responses to the reviewers' comments are as follows:
Response to reviewer's comments:
Reviewer 1
Comment 1
Table 1 should be improved, reporting data of mortality of principals trials and multi center international registries published since yet and reporting an extra column with ongoing trials.
Author's response: We have revised Table 1 accordingly, and Table 2 shows the ongoing trials. (Please see Table 1 and Table 2)
Comment 2
When reporting principals trials for MitraClip, I would specify that the EVEREST trial was a pioneering trial that selected mostly DMR patients with lower risk than in real-world practice. Similarly, I would specify that recent COAPT and MITRA-FR trials were conducted just in FMR patients.
Author's response: We have mentioned in the paper the differences in the selection of patients for the EVEREST trial, the COAPT trial and the MITRA-FR trial. (Please see line 121-122 and 128)
Comment 3 and 4
As regards future directions, I would underline also the impact of different etiology of MR upon outcome, including the new definitions of atrial functional mitral regurgitation (of which we have preliminary experience, mostly with MitraClip device). This aspect could be underlined also for other devices, for example transcatheter annular devices.
In Discussion, the authors mainly focused upon mitral valve replacement and discussion regarding mitral valve repair is really poor. Moreover, the Discussion paragraph starts with citation of COAPT trial and the challenges regarding patients with secondary MR, who are not the main or only focus of the paper (what about DMR?). I would change the cut of discussion and I would expand this section citing the challenges that may occur with mitral valve repair, the differences (also technical) that we can meet with different etiologies and the process of choosing the different devices (in this case, are them competitive or complementary? And when or how?). In this regard, a graphical algorithm may be suitable and of good impact to the readers.
Author's response: We reoriented the discussion to the differences encountered in TMVr with different etiologies and challenges, and mentioned the uniqueness of atrial functional mitral regurgitation and possible therapeutic approaches. Graphical algorithms have been added. (Please see line 413-433 and Figure 3)
Comment 5
Most of the aspects regarding TMVR have been addressed in a recent review that i suggest to cite: Luca Testa, et al. "Transcatheter mitral valve replacement in the transcatheter aortic valve replacement era." Journal of the American Heart Association 8.22 (2019): e013352.". I would add also in this case a summary Table with all the ongoing studies regarding TMVR as reported in this paper that I suggest.
Author's response: We have added a table to describe the ongoing TMVR studies and related citations. (Please see Table 5)
Comment 6
Initial clinical experience with HighLife have been published. I suggest to add the paper "Marco Barbanti, et al. "Transcatheter mitral valve implantation using the HighLife system." JACC: Cardiovascular Interventions 10.16 (2017): 1662-1670".
Author's response: We have described the outcomes of the initial clinical experience. (Please see line 387-388)
Comment 7
I have a comment regarding the sentence "TMVr, on the other hand, can only fit a limited number of patients and can be only performed by highly-experienced operators". My general opinion (but I admit that it can be absolutely personal) is that TMVR is more technically challenging that TMVr. Moreover, anatomical suitability for TMVR can be more difficult in some cases rather than TMVr. I would not be so categorical. And finally: regarding the title question (competitive or complementary?), I am not really sure that the authors answered to the question.
Author's response: We have added some discussion about the impact of institutional and operator experience on the TMVr procedure. As far as the current study results are concerned, both TMVr and TMVR have challenges to overcome, so they are complementary. (Please see line 434-436)
Thank you for your careful review. We really appreciate your efforts in reviewing our manuscript. Your careful review has helped to make our review clearer and more comprehensive. Thanks again!

Reviewer 2 Report
The manuscript entitled “Transcatheter Mitral Valve Repair or Replacement:Competitive or Complementary?” describes the roles of TMVr versus TMVR in the near future addressing the main devices for repair and for replacement.
The manuscript is well presented and the topic is interesting. Please find some comments below in order to improve the overall quality of the paper.
- The topic is quite complex and especially for the replacement data are still scarce. However, it is important to discuss the very high selection failure of TMVR (reaching up to 70-80%) which limits significantly such technology.
- LVOTO is probably the main cause for selection failure. In case of valve in MAC it reaches 30% of mortality according to data by Yoon SH, et al Outcomes of transcatheter mitral valve replacement for degenerated bioprostheses, failed annuloplasty rings, and mitral annular calcification. Eur Heart J. 2019 Feb 1;40(5):441-451.
- Please emphasize the importance of LVOTO.
- When discussing TMVR, it would be advisable to distinguish between TMVR in native mitral valve, valve in valve, valve in ring and valve in MAC. The outcomes are quite different and cannot be generalized. For this you may cite and report data from Russo G, Gennari M, Gavazzoni M, Pedicino D, Pozzoli A, Taramasso M, Maisano F. Transcatheter Mitral Valve Implantation: Current Status and Future Perspectives. Circ Cardiovasc Interv. 2021 Sep;14(9):e010628. This manuscript presents also an treatment algorhythm that might be useful.
- When comparing TMVR and TMVr it is important also to mention the role of centre and operator’s experience. In this regard it is important to cite the TVT manuscript (Chhatriwalla AK, Vemulapalli S, Holmes DR Jr, Dai D, Li Z, Ailawadi G, Glower D, Kar S, Mack MJ, Rymer J, Kosinski AS, Sorajja P. Institutional Experience With Transcatheter Mitral Valve Repair and Clinical Outcomes: Insights From the TVT Registry. JACC Cardiovasc Interv. 2019;12(14):1342-1352.) and also the manuscript by Gavazzoni M, Taramasso M, Zuber M, Russo G, Pozzoli A, Miura M, Maisano F. Conceiving MitraClip as a tool: percutaneous edge-to-edge repair in complex mitral valve anatomies. Eur Heart J Cardiovasc Imaging. 2020;21(10):1059-1067.
Author Response
Dear editors and reviewers:
Thank you very much for your careful review and your very constructive suggestions with regard to our manuscript.
We have made the necessary corrections. Revised portions are marked in red in the paper. The main corrections in the paper and the responses to the reviewers' comments are as follows:
Response to reviewer's comments:
Reviewer 2
Comment 1 and 2
The topic is quite complex and especially for the replacement data are still scarce. However, it is important to discuss the very high selection failure of TMVR (reaching up to 70-80%) which limits significantly such technology.
LVOTO is probably the main cause for selection failure. In case of valve in MAC it reaches 30% of mortality according to data by Yoon SH, et al Outcomes of transcatheter mitral valve replacement for degenerated bioprostheses, failed annuloplasty rings, and mitral annular calcification. Eur Heart J. 2019 Feb 1;40(5):441-451.Please emphasize the importance of LVOTO.
Author's response: We have discussed the severity and solutions of LVOTO and mentioned a method for calculating neo-LVOT that may identify potentially suitable patients. In addition, the application of some imaging techniques may improve the high selection failure of TMVR. (Please see line 466-475 and line 540-558)
Comment 3
When discussing TMVR, it would be advisable to distinguish between TMVR in native mitral valve, valve in valve, valve in ring and valve in MAC. The outcomes are quite different and cannot be generalized. For this you may cite and report data from Russo G, Gennari M, Gavazzoni M, Pedicino D, Pozzoli A, Taramasso M, Maisano F. Transcatheter Mitral Valve Implantation: Current Status and Future Perspectives. Circ Cardiovasc Interv. 2021 Sep;14(9):e010628. This manuscript presents also an treatment algorhythm that might be useful.
Author's response: We have categorized TMVR and discussed the results in different cases. (Please see line 252-272)
Comment 4
When comparing TMVR and TMVr it is important also to mention the role of centre and operator's experience. In this regard it is important to cite the TVT manuscript (Chhatriwalla AK, Vemulapalli S, Holmes DR Jr, Dai D, Li Z, Ailawadi G, Glower D, Kar S, Mack MJ, Rymer J, Kosinski AS, Sorajja P. Institutional Experience With Transcatheter Mitral Valve Repair and Clinical Outcomes: Insights From the TVT Registry. JACC Cardiovasc Interv. 2019;12(14):1342-1352.) and also the manuscript by Gavazzoni M, Taramasso M, Zuber M, Russo G, Pozzoli A, Miura M, Maisano F. Conceiving MitraClip as a tool: percutaneous edge-to-edge repair in complex mitral valve anatomies. Eur Heart J Cardiovasc Imaging. 2020;21(10):1059-1067.
Author's response: We have added a discussion of the impact of institutional and operator experience on TMVr. (Please see line 434-436)
Thank you for your careful review. We really appreciate your efforts in reviewing our manuscript. Your careful review has helped to make our review clearer and more comprehensive. Thanks again!

Reviewer 3 Report
I would like to congratulate the authors for this rather nice overview of TMVr(R) techniques. The topic is carefully selected, very appealing, and most importantly, important for clinical practice. There is a lack of similar papers within the field that summarize all the aspects and possibilities of transcatheter modalities.
Please correct the line 84 (Thethe), and the line 273 (first mention of the abbreviation MAC).
Otherwise, no major nor minor comments. Very happy with the quality of the paper.
Author Response
Dear editors and reviewers:
Thank you very much for your careful review and your very constructive suggestions with regard to our manuscript.
We have made the necessary corrections. Revised portions are marked in red in the paper. The main corrections in the paper and the responses to the reviewers' comments are as follows:
Response to reviewer's comments:
Reviewer 3
Comment 1 Please correct the line 84 (Thethe), and the line 273 (first mention of the abbreviation MAC).
Author's response: We have revised the problems accordingly. (Please see line 114, 270 and 319)
Thank you for your careful review. We really appreciate your efforts in reviewing our manuscript. Your careful review has helped to make our review clearer and more comprehensive. Thanks again!

Round 2
Reviewer 1 Report
Dear Authors,
thank you for having answered to all my questions. I think the manuscript has significantly improved, both in the content and images/tables. Readability has increased too.
Just a minor comment:
As regards atrial functional mitral regurgitation, recent works regarding percutaneous treatment in this setting have been published and should be included:
1. Popolo Rubbio A, et al. Transcatheter edge-to-edge mitral valve repair in atrial functional mitral regurgiation: insights fromt the multi-center MITRA-TUNE registry; International Journal of Cardiology (2022), 349: 39-45
2. Rottandler D, et al. Mitral Valve edge-to-edge repair versus indirect mitral valve annuloplasty in atrial functional mitral regurgitation; Catheterization and Cardiovascular Interventions (2022), 99: 1839-1847
Author Response
Dear editors and reviewer:
Thank you very much for your careful review and your very constructive suggestions with regard to our manuscript.
We have made the necessary corrections according to your minor revision. Revised portions are marked in red in the paper. The main corrections in the paper and the responses to the reviewers' comments are as follows:
Response to reviewer's comments:
Comment 1
As regards atrial functional mitral regurgitation, recent works regarding percutaneous treatment in this setting have been published and should be included:
Author's response: Thank you very much for your suggestion. The relevant paper has been included. (reference No. 77 and 78)